# A Novel Hair Restoration Technology Counteracts Androgenic Hair Loss and Promotes Hair Growth in A Blinded Clinical Trial

**DOI:** 10.3390/jcm12020470

**Published:** 2023-01-06

**Authors:** Dominik Thor, Andrea Pagani, Julia Bukowiecki, Khosrow S. Houschyar, Stig-Frederik T. Kølle, Saranya P. Wyles, Dominik Duscher

**Affiliations:** 1Tomorrowlabs GmbH, 1010 Vienna, Austria; 2Department of Orthopedics, Traumatology and Hand Surgery, Hospital of Bolzano-SABES, Lorenz-Böhler-Straße 5, 39100 Bolzano, Italy; 3Department of Plastic Surgery, SANA Clinic Düsseldorf Gerresheim, Gräulingerstraße 120, 40625 Düsseldorf, Germany; 4Department of Dermatology and Allergology, University Hospital Aachen, 52074 Aachen, Germany; 5Department of Plastic Surgery, Cerix Private Hospital, 1150 Copenhagen, Denmark; 6Department of Dermatology, Center for Regenerative Biotherapeutics, Mayo Clinic, Rochester, MN 55902, USA; 7Department of Hand, Plastic, Reconstructive and Burn Surgery, BG-Unfallklinik Tuebingen, University of Tuebingen, Schnarrenbergstraße 95, 72076 Tübingen, Germany

**Keywords:** androgenic alopecia, dermal papilla, HIF-1α, hair loss, hypoxia-inducible factor, clinical trial

## Abstract

Androgenic alopecia (AGA) is a genetically predetermined condition that occurs as a result of stepwise miniaturization of the dermal papilla. During this process, the hair follicle suffers from increasing malnutrition and eventually dies, causing progressive hair loss. We recently highlighted that HIF-1α modulation may counteract hair loss. Here, we aim to demonstrate the positive influence of Tomorrowlabs HIF strengthening factor [HSF] hair restoration technology on hair biology in a monocentric blinded clinical trial over a total period of 9 months. A trial with 20 subjects (4 female and 16 male) and once-daily application of [HSF] hair restoration technology to the scalp was conducted. To assess the tolerability and efficacy of [HSF], testing included dermatological assessment, determination of hair loss by counting after combing, macro images of the head and TrichoScan evaluation of hair density as well as the proportion of anagen hair versus telogen hair. The clinical data show Tomorrowlabs [HSF] hair restoration to be safe and effective to counteract AGA. The use of Tomorrowlabs [HSF] hair restoration resulted in improvements in the clinical parameters of hair quality such as thickness (+7.2%), hair density (+14.3%) and shine and elasticity (+20.3%) during the test period. The effectiveness of the test product was further determined by a significant reduction in hair loss of an average of 66.8% in treatment-responsive subjects after 6 months and an increase in hair growth reaching up to 32.5%, with an average percentage change of 8.4% in all participants and 10.8% in the responsive patients (85% of the study cohort) after 9 months on TrichoScan evaluation. The hair growth cycle was harmonized with the result of an average anagen hair percentage increase of +8.0% and telogen hair percentage reduction of −14.0% shown in the test area. Applicable for both sexes in an alcohol-free formulation, beneficial to scalp health and free of complications or side effects, this novel product provides objectively measurable results counteracting hair loss paired with an improved look and feel of the hair.

## 1. Introduction

The skin is the largest and most functionally versatile human organ. By separating the body from internal and external environmental factors, it represents a key element of the protection of the body’s homeostasis. The skin consists of the epidermis, the most superficial layer; the dermis, which is the middle layer and the subcutis, the innermost of the three layers [1]. The epidermis, in turn, is made up of five layers and consists of 90% keratinocytes. The superimposed layers follow from the outside inwards: stratum corneum, lucidum, granulosum, spinosum and basale [2].

In the past, skin and hair were considered protective and accessory elements; nowadays, they are acknowledged to be functional units. In addition to skin homeostasis, regeneration and repair, hair has key functions such as thermoregulation, skin protection, sebum production and apocrine sweat and pheromone secretion as well as social interaction [3]. At the base of the hair, the hair follicle (HF) is mainly composed of epithelial and dermal papilla cells (DPCs). DPCs belong to the mesenchymal compartment of the hair bulb and are one of the main regulators of the hair cycle [4]. Even the dermal papilla niche has a crucial role in the hair growth process and regeneration cycle, alternating between phases of active growth (anagen), regression (catagen), and relative “quiescence” (telogen) [5]. Androgenic alopecia (AGA), or male- and female-pattern hair loss, is a stepwise and genetically programmed clinical disorder affecting the majority of the male and almost half of the female population [6]. AGA occurs as a result of the weakening of dermal papillae, a spheroid-shaped structure that provides nutrition and oxygen to the HF. Although the etiopathogenesis is not yet fully understood, AGA is associated with a progressive shortening of the anagen phase (hair growth and production) and an elongation of the telogen phase (hair follicle dormancy) [7]. In addition, during hair loss, the hair follicle undergoes a particular “miniaturization” process during which the hair becomes thinner and weaker. The downsizing of the hair shaft is associated with a reduction in the size of the dermal papilla (DP), leading to consequential hair loss. The size of the DP is proportional to the diameter of the hair shaft [8]. If untreated, the HF loses nutrition over time, triggering the process of miniaturization, and is thus unable to re-enter the anagen phase and eventually dies.

There is a key signaling pathway able to regulate hair development, repair and regeneration. Our group recently confirmed the importance of the HIF (Hypoxia-Inducible Factor) pathway as a paradigm of molecular signaling governing both skin and hair regeneration and, in particular, modulating the shape and size of the dermal papilla [9,10,11]. In particular, hair growth has some similarities to wound healing, requiring a highly coordinated interplay between tissue formation and cell growth and migration. The Hypoxia-Inducible Factor 1α (HIF-1α) drives neovascularization and collagen and elastin production during wound healing. AGA has been linked to a lack of blood vessels and nutrient supply in the hair bulb. Hence, HIF stimulation can modulate both neovascularization and regeneration because DPCs are reactive to hypoxia.

At present, several topical drugs claim to prevent, stop and even partially reverse hair loss. Among these, minoxidil is the current market leader. Formerly taken orally as a treatment for hypertension with the side effect of hypertrichosis, it is currently available as a topical 2% and 5% tonic or foam that has to be applied twice a day. Although the cause of its effect on hair growth has not yet been established, the general consensus gives credit to its vasodilating properties.

In particular, minoxidil’s molecular mechanism of action is strictly connected with the HIF pathway. It is widely known that the prolyl-hydroxylase 2 (PHD-2)-mediated hydroxylation of HIF-1α causes its rapid degradation. Blocking PHD-2 leads to HIF-1α survival and hypoxia-response element gene (HRE) transcription such as VEGF, CCL5 and Endothelins ET–1 and ET–2 [12]. Accordingly, minoxidil stimulates HIF and all its angiogenic components. The anagen phase of the hair cycle is characterized by increased vascularization [13]; hence, neovascularization due to minoxidil triggers dormant hair follicles to re-enter the anagen phase. On the other hand, the effect of minoxidil can be abrogated by the addition of ascorbate, implying that it is a competitive inhibitor of PHD at the binding site of ascorbate, a co-factor of PHD [14]. Given these mechanistic insights, it is not surprising that the recently introduced substance Stemoxydine^®^ (proprietary to L’Oreal, Paris, France) is also a PHD competitive inhibitor [15]. Inducing hypoxia-like signaling via HIF-pathway modulation is the mechanism of action behind the described hair-biology-enhancing effect of Stemoxydine^®^ 5% hydro-alcoholic lotion [16].

HIF strengthening factor [HSF] acts as an iron chelator, and iron is a crucial co-factor of PHD, so by chelating iron, there is none left to activate PHD. The inactivation of PHD, again, leads to high levels of undegraded HIF-1α, which is able to bind and dimerize with HIF-1β to build HIF-1. The transcription of the HRE leads to strong pro-regenerative and pro-angiogenic action. Applied topically, the small active ingredient [HSF] is able to penetrate the skin [17]. Because the decrease in blood flow to the HF is a dominant element of hair loss, the use of angiogenic pharmaceutical agents is a logical path to preventing hair loss.

We previously demonstrated the positive influence of HIF strengthening factor [HSF] hair restoration technology on hair biology in vitro [10]. Using the hanging-drop method, DPCs were treated with traditional substances against AGA, such as minoxidil and caffeine, compared to HIF-1α-stimulating agents in vitro. DPC proliferation within the spheroid structure was highest in the HSF-treated DPCs [10].

The purpose of this report is:

Objective assessment of the safety and tolerability of Tomorrowlabs [HSF] hair restoration products in humans.

Objective measures of hair look and feel after Tomorrowlabs [HSF] hair restoration treatment.

Objective examination of the capacity of Tomorrowlabs [HSF] hair restoration to counteract AGA in vivo by reducing hair loss and promoting hair growth.

## 2. Methods

The product tested in this application study was used on the intended application area over 9 months. A risk analysis of the components of the test product was carried out before the beginning of the study. All available information was systematically evaluated to identify potential dangers and risks.

According to dermatological test criteria, the main goal of this work was to examine the tolerability and effectiveness of the [HSF] hair restoration product. In the beginning, the health and integrity of the integument of the subjects were examined. If a condition requiring medical treatment existed, the subject was excluded. There was also an informative discussion in which the study design, the study conditions and the rights and obligations of the test persons within the framework of the study were explained to the test persons by the supervising study nurse or the supervising dermatologist. The test persons were only included in the study if they did not show any pathological skin changes in the application zone, either signed the declaration of consent themselves or had it signed by a legal guardian and met the inclusion and exclusion criteria. During the study, the subjects could consult the supervising nurse and the supervising dermatologist regarding objective or subjective skin changes. According to the schedule, all necessary clinical and dermatological examinations took place. In order to keep fluctuations due to external influences such as room temperature and relative humidity as low as possible, the measurements were always carried out under the same physical environmental conditions when the patient was rested (20 °C, humidity 40–60%).

### 2.1. Patient Recruitment and Study Conduct

The data analysis and study conduct were based on the principles of GCP and were in agreement with the Helsinki Declaration. Patients were all in good health, on average 42.55 years old (SD 7.02), with genetically caused hair loss (Androgenic alopecia, AGA) in the hair growth stage according to Norwood III–IV (male test persons) or Ludwig–Level 1 (female subjects) qualified for inclusion (Figure 1). Pregnant or lactating women and patients with allergies to hair care products were excluded from the study. Patients who had received platelet-rich plasma (PRP) or laser or other scalp-altering treatments within 6 months of the start of the study were not included.

After signing the informed consent, 4 female subjects (mean age 46.50, SD 6.66, range 37–52 years) and 16 male subjects (mean age 41.56, SD 6.96, range 31–57 years) were included in this study.

[HSF] hair restoration product was applied by blinded users once daily to damp hair after a shower and massaged in at the areas of hair loss. The first application of the product was performed under the supervision of a study nurse.

### 2.2. Dermatological Assessment of Skin Tolerability and Hair Look and Feel

Demographic data, scalp status and clinical history were collected and analyzed during an initial phase of screening. Tomorrowlabs [HSF] hair restoration tolerability was deeply evaluated by a board-certified dermatologist. Treated areas were examined under constant light conditions. In the dermatological assessment after the in vivo touching evaluation, various parameters were determined visually and by touching the skin/hair. This assessment was carried out by a trained assessor using a visual analog scale to evaluate the relevant parameters [18]. The scale was defined from “no intensity/excellent condition” to “maximum intensity/poor condition”. The individual intensities are displayed numerically by measuring the analog scale (values 0.0 to 100.0).

### 2.3. Determining Reduced Hair Loss

Hair loss after combing was determined according to an accepted method described in the literature [19]. After the hair had been washed, the still-damp hair was thoroughly combed for 60 s with a standardized comb. The teeth of the comb, which was 15 cm long, were separated by 1 mm on one half and by 2 mm on the other half of the comb. The hair that had fallen out was counted by a qualified study nurse.

### 2.4. Determining Increased Hair Growth

For a visual assessment of the affected skin areas, digital images (macro images) were taken with the Nikon D200 and the AF-S Micro Nikkor lens (focal length 60 mm; speed 1:2.8; type G ED glass; exposure time 1:200; aperture F/ 22; ISO 200). For each subject, one picture was taken before the start of the application and one picture after the end of the application.

The TrichoScan HD method (DermoScan GmbH) evaluates hair growth objectively. One measurement area is determined on the scalp of each subject. The pre-treatment of the scalp was carried out as follows: A measuring area along the transition zone between the alopecia and the regular hairy scalp is selected for each subject. The subsequent measurement area is defined with the aid of a perforated mask. The hair is threaded through the perforated mask and roughly shortened with scissors. The shortened hair is shaved to an even length of 0.8 mm with the Moser shaver (TrichoScan Edition). Without applying any pressure, the razor is guided at a 90° angle to the scalp. Hair is then colored two days after shaving; the hair color (Goldwell Topchic 2N) is applied to a wooden spatula. The same amount of developer (Creme oxyd) is added to the hair color (1:1 mixture). Hair color and developer are mixed together in order to obtain a product with a creamy consistency. The coloring mass is applied to the measurement area (scalp) of the test subject and acts for 15 min. After the exposure time, the stain is roughly removed with a swab, and the measuring area has to be cleaned with an alcoholic tincture (e.g., Kodan Spray). Image files are recorded with the camera handpiece of the TrichoScan HD without any air bubbles and surrounding hair. To take the pictures, the measuring area is moistened with a particular Kodan Spray. Then, the parameters of hair density [1/cm^2^], number of hairs, anagen hairs [%] and telogen hairs [%] are determined using TrichoScan HD software.

### 2.5. Statistical Analysis

Research subjects were measured on five characteristics prior to the use of Tomorrowlabs [HSF] hair restoration. These were: thickness, hair density, shine and elasticity, hair loss and hair growth. At the end of the test period, the subjects were measured again. The before and after values of each characteristic were compared to assess the effectiveness of Tomorrowlabs [HSF] hair restoration. For all characteristics, with the exception of hair loss, the test period was 9 months. For hair loss, the test period was 6 months. The Shapiro–Wilk test showed that the subject data on thickness, hair density, shine and elasticity and hair growth were normally distributed (*p* > 0.05). Therefore, comparisons were assessed using paired-sample t tests. The Shapiro—Wilk test indicated that hair loss was non-normally distributed (*p* < 0.05). As a result, mean differences were assessed using the Wilcoxon signed-rank test. For all tests, a resulting p value of less than 0.05 was considered significant.

## 3. Results

### 3.1. Dermatological Assessment Proves Tolerability of [HSF] Hair Restoration

All clinical and dermatological examinations were performed according to clinical-dermatological assessment criteria. All subjects reported healthy skin in the treated area before, during and after the application study. No pathological skin changes were detected. Test interruptions or even dermatological treatments were not carried out in any case. The preparation mentioned was very well tolerated and did not lead to relevant skin changes in any of the test persons. The results are presented in Table 1 and discussed below.

### 3.2. Dermatological Assessment Demonstrates Highly Significant Improvement in Hair Quality via [HSF] Hair Restoration

The Tomorrowlabs [HSF] hair restoration results were evaluated by board-certified dermatologists according to a standardized protocol. The use of Tomorrowlabs [HSF] hair restoration resulted in highly significant improvements in the clinical parameters of hair quality such as improvements in thickness of 7.2% (*p* < 0.001), hair density of 14.3% (*p* < 0.001) and shine and elasticity of 20.3% (*p* < 0.001) during the 9-month test period (Figure 2).

### 3.3. Tomorrowlabs [HSF] Hair Restoration Significantly Reduces Hair Loss

The mean delta of hair loss across all study participants after 6 months of study was −30.36% (*p* < 0.05) (Figure 3A). At this time, 80% of the study participants (16 out of 20) observed a reduction in hair loss. In these 16 responding subjects, a mean reduction in hair loss of 66.8% (*p* < 0.05) was achieved, corresponding to 37 hairs lost on average after 6 months (Figure 3B). Interestingly, we found that for the top 20% of the study participants suffering from the most active hair loss as reflected by hair count at t(0), it was still possible to achieve a dramatic reduction in hair loss with an average improvement of more than 60% after 6 months (*p* < 0.05), effectively reducing their hair shedding from an average of 275 hairs a day to 137 and almost normalizing their hair turnover to a physiological range (up to 100 hairs lost per day) (Figure 3C). The average hair loss in the study cohort after 6 months corresponds well with the mean hair loss rate in healthy patients, which was found to be approximately 40 according to the literature (20). The maximum reduction in hair loss achieved in the study population was 100%.

### 3.4. Tomorrowlabs [HSF] Hair Restoration Highly Significantly Increases Hair Growth

After 9 months, hair growth improvement was visible macroscopically in most of the study participants (Figure 4A), and the difference between the before and after measurements on the TrichoScan was significant (*p* < 0.01). The TrichoScan evaluation was carried out on a test area on the scalp. The hair density [1/cm^2^], the proportion of anagen hair [%] and the proportion of telogen hair [%] were determined to assess the influence of the Tomorrowlabs [HSF] hair restoration products on hair growth. After 9 months of treatment, an increase in hair growth of up to 32.5% (10.76% on average in responsive patients, *p* < 0.001) was measured (Figure 4B). In only 3 patients, there was no improvement in hair density, which equals an 85% responder rate in our study cohort.

### 3.5. Tomorrowlabs [HSF] Hair Restoration Harmonizes the Hair Cycle

The TrichoScan evaluation revealed a harmonization of the hair growth cycle with results of an average anagen hair percentage increase of +8.0% and telogen hair percentage reduction of −14.0% shown in the test area (Figure 5).

## 4. Discussion

Many products, especially cosmetics, consumer goods and medical devices, are in daily contact with the skin, often for long periods of time. Good tolerability is, therefore, a prerequisite for the use of these products. Because alternative test methods such as animal experiments are prohibited in Europe, and the results of cell culture experiments can only be transferable to humans to a limited extent, tests on humans carried out under medical supervision are currently necessary from an ethical and scientific point of view.

This report aims to determine the clinical safety and efficacy of Tomorrowlabs [HSF] hair restoration products in the context of AGA. From a dermatological but even psychological perspective, while hair shedding counts are particularly important, growth rate and anagen/telogen ratio are of secondary importance [20]. Healthy people have between 80,000 and 120,000 hairs. In healthy individuals, approximately 10–20% of the hair follicles are in the resting stage (telogen) at any given time, which means that they are ready for the strand to fall out. In our cohort, we were able to reduce the amount of hair in the telogen phase from 31.8% at the beginning of our study to 26.9% at the end of the study period, effectively rebalancing the hair follicle towards a more regenerative status.

Using a standardized combing test is a widely accepted method to assess the number of hairs shed [19]. The 60-s hair count represents an objective and practical tool for monitoring hair-shedding conditions. The low intrapatient variability demonstrates that dependable results over an extended period of time are obtainable [19]. The mean hair-loss rate of this method in healthy patients was found to be approximately 40 [21]. In our study cohort of AGA patients, we were able to reduce mean hairs lost via treatment with Tomorrowlabs [HSF] hair restoration from 69 at T(0) to 38 after 6 months, which represents a restoration of normal hair homeostasis. This is a best-in-class result not achieved by any other product in the market to the best of our knowledge. In our cohort, we observed an 80% responder rate, where either a reduction in hair loss or a halting of the progression of hair loss was achieved. Interestingly, we found that for the top 20% of the study participants suffering from the most active hair loss reflected by hair count at t(0), we were still able to achieve a dramatic reduction in hair loss with an average improvement of more than 60% after 6 months. This is a strong indicator that Tomorrowlabs [HSF] hair restoration is also able to achieve the best results when applied for the condition of aggressive hair loss.

The traditional goals of a quality hair-growth product are to heal damaged hair follicles and prolong their growth phase. Hence, any successful treatment should stop or reverse the hair follicle miniaturization process and increase the number of terminal hair follicles while reducing vellus hair counts and increasing the number of anagen hairs. However, anagen phase prolongation is also the main reason for the shedding phenomenon associated with minoxidil. HIF modulators/PHD inhibitors have been shown to execute their beneficial effects on hair biology via a reduction in the kenogen phase, the latency period required for new hair growth to engage [15,16]. This enables a positive influence on hair-cycle dynamics while preventing abrupt shedding when the treatment is beginning or discontinued.

An ideal hair-growth product is not only beneficial for hair quality and count but also makes your existing hair healthier and nourishes the skin beneath it. One of the major issues related to treatment with minoxidil is that minoxidil is beneficial for hair growth but detrimental to scalp health. Minoxidil is associated with a specific loss of lysyl hydroxylase activity, and this loss can be reversed after removing minoxidil [22]. By acting at the transcriptional level, minoxidil inhibits lysyl hydroxylase synthesis [23], which is key for collagen stabilization. In addition, minoxidil treatment is also associated with the inhibition of cell proliferation [24]. This effect happens because of DNA synthesis inhibition in skin cells. Because collagen is the major product of fibroblast activity and lysyl hydroxylase catalyzes a pivotal reaction in collagen biosynthesis, minoxidil can be harmful to fibroblast functionality and, consequentially, skin homeostasis [25].

This is further aggravated by the fact that minoxidil treatment requires patients to apply the drug to the scalp every 12 h to achieve hair growth because of its pharmacokinetic properties [14]. One important factor of minoxidil in this context is its insolubility in water; it is dissolved into alcohol and sold as such. Applying it twice daily leads to irritation of the skin, because alcohol as a vehicle is deeply dehydrating [26]. This set of negative characteristics is likely to be shared by newer compound developments such as Stemoxydine^®^ (proprietary to L´Oreal, France). Stemoxydine^®^ is a PHD competitive inhibitor [16] and, therefore, hypoxia-like signaling via HIF-pathway modulation is the mechanism of action. However, not only does Stemoxydine^®^ share the HIF pathway as a common denominator of efficacy with minoxidil, but its delivery is also based on a hydro-alcoholic lotion potentially harmful to the scalp [16]. In stark contrast, Tomorrowlabs [HSF] hair restoration products are formulated without alcoholic penetration enhancers, enabling powerful HIF modulation without skin toxicity. This results in the complete absence of complications and adverse events in the clinical assessment, while competitors such as minoxidil cause considerable skin irritation in a considerable number of customers. For Stemoxydine^®^, no peer-reviewed data on adverse reactions are available. The safety advantage of [HSF] is also accompanied by increased usability. With a different pharmacological mechanism underlying HIF modulation via [HSF], less frequent application yields efficacy. In our study collective, we are able to demonstrate significantly reduced hair loss and increased hair growth upon application of Tomorrowlabs [HSF] hair restoration products once daily.

## 5. Conclusions

Taken together, our clinical data clearly demonstrate Tomorrowlabs [HSF] hair restoration to be safe and effective to counteract AGA. Applicable for both sexes and free of complications and side effects, this novel product provides objectively measurable results. This study shows that Tomorrowlabs [HSF] hair restoration technology and products are water-soluble alternatives to the market leaders in hair-loss treatments, minoxidil and caffeine. With proven efficacy in a clinical assessment, [HSF] demonstrates significantly reduced hair loss and increased hair growth in humans. [HSF] treatment is associated with beneficial effects on hair quality and scalp health. These favorable characteristics are further supported by its satisfactory outcomes and the absence of negative skin reactions. Given the promising results, the [HSF] represents a true innovation in AGA treatment.

## Figures and Tables

**Figure 1 jcm-12-00470-f001:**
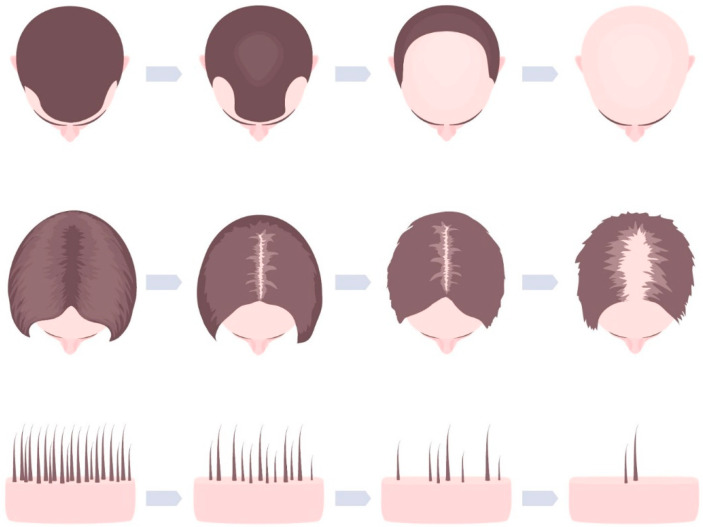
Typical pattern of androgenic hair loss in males (Norwood scale, **above**) and females (Ludwig scale, **below**).

**Figure 2 jcm-12-00470-f002:**
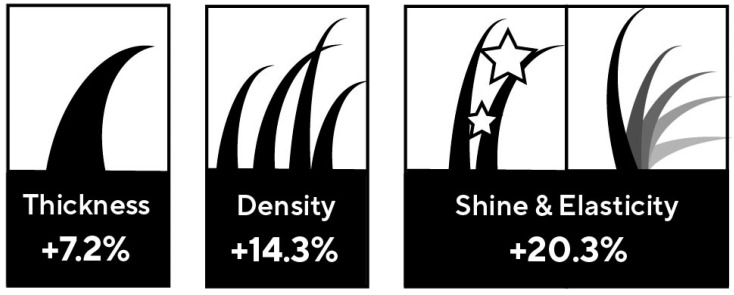
Hair quality was improved significantly by using Tomorrowlabs [HSF] hair restoration treatment. The Tomorrowlabs [HSF] hair restoration results were evaluated by board-certified dermatologists according to a standardized protocol. The use of Tomorrowlabs [HSF] hair restoration resulted in improvements in the clinical parameters of hair quality such as improvements in thickness of 7.2% (*p* < 0.001), hair density of 14.3% (*p* < 0.001) and shine and elasticity of 20.3% (*p* < 0.001) during the 9-month test period.

**Figure 3 jcm-12-00470-f003:**
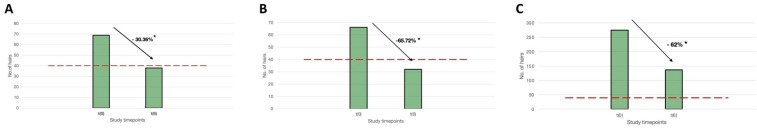
Tomorrowlabs [HSF] hair restoration significantly reduces hair loss. (**A**) The results of hair loss assessment via combing show a significant decrease in hair loss in the study group. The mean delta of hair loss across all study participants was −30.36%, resulting in an average reduction of 38 hairs lost at t(6) vs. 69 hairs lost at T(0) (* *p* < 0.05). (**B**) For responsive participants, an average hair loss reduction of 66.8% was measured, corresponding to 37 hairs lost on average at t(6) (* *p* < 0.05). (**C**) Interestingly, we found that for the top 20% of the study participants suffering from the most active hair loss reflected by hair count at t(0), it was still possible to achieve a dramatic reduction in hair loss with an average improvement of more than 60% at t(6) (* *p* < 0.05). Average physiological hair loss depicted by red dashed line.

**Figure 4 jcm-12-00470-f004:**
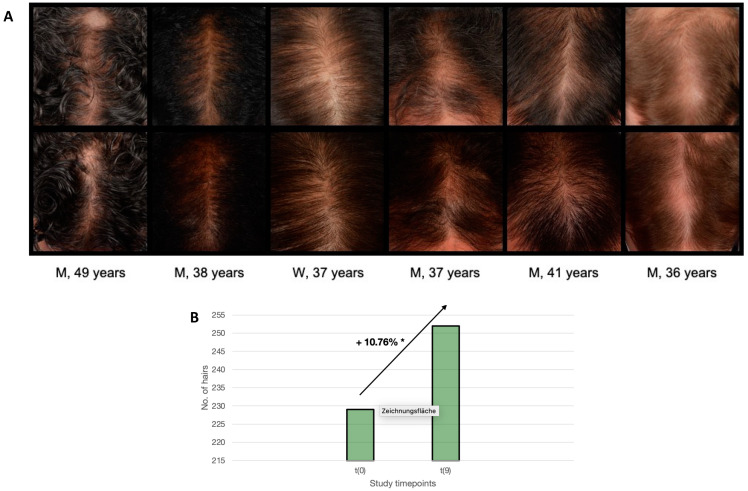
Tomorrowlabs [HSF] hair restoration significantly increases hair growth. (**A**) Clinical examples of macroscopic hair growth improvement in the study period. (**B**) After 9 months of treatment, an increase in hair growth of 10.76% on average in responsive patients was measured (* *p* < 0.001).

**Figure 5 jcm-12-00470-f005:**
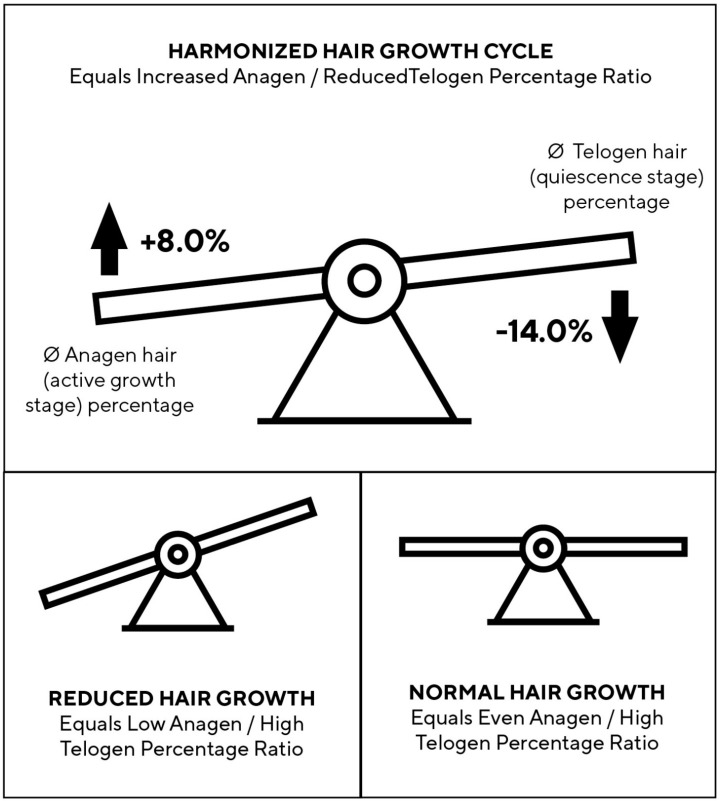
Tomorrowlabs [HSF] hair restoration harmonizes the hair growth cycle. The hair growth cycle was harmonized, with results of an average anagen hair percentage increase of +8.0% and telogen hair percentage reduction of −14.0% shown in the test area after 9 months of treatment.

**Table 1 jcm-12-00470-t001:** Comparison before and after use of Tomorrowlabs [HSF] hair restoration.

Variable	*N*	Before	After	*p* Value
Thickness, mean (SD)	20	57.89 (16.43)	53.94 (16.55)	<0.001
Hair density, mean (SD)	20	59.39 (18.76)	51.68 (20.22)	<0.001
Shine and elasticity, mean (SD)	20	64.84 (19.46)	52.67 (19.24)	<0.001
Hair loss, mean (SD)	20	68.55 (127.49)	38.25 (81.32)	<0.05
Hair loss, responders, mean (SD)	16	77.29 (33.22)	36.76 (21.29)	<0.05
Hair loss, top 20%, mean (SD)	4	275.25 (174.97)	137.25 (152.38)	<0.05
Hair growth, mean (SD)	20	232.67 (42.21)	250.20 (40.60)	<0.01
Hair growth, responders, mean (SD)	17	229.45 (44.77)	252.27 (42.84)	<0.001

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
