# Peer review of "A Novel Hair Restoration Technology Counteracts Androgenic Hair Loss and Promotes Hair Growth in A Blinded Clinical Trial"

_jcm, 2023, doi:10.3390/jcm12020470_

Round 1
Reviewer 1 Report
Improve the statistical part and report the results in a readable way.
"Insert average age and standard deviation instead of 30 and 60 years".
"Two-tailed Student's 175 t test was used to perform a direct comparison between the timepoints and results were considered significant at p ≤ 0.05"
This sentence is not clear what it refers to. insert a paragraph from the statistical analysis
"The use of Tomorrowlabs [HSF] hair restoration resulted in an improvement in the clinical parameters of hair quality such as thickness of 7.2%, hair density of 14.3%, shine and elasticity of 20.3% during the test period
(Figure 2)."
prove it with a statistical test
Redo the figures by removing the title and inserting a caption for each and indicating the p-value and the test used. the figures are of low quality and clarity
The caption in Figure 3 shows panels A, B, C, D but the letters are not shown in the figure.
"Average physiological hair loss depicted in red dashed line. (*p < 0.05)."
I do not understand what this means
In the bar plot in figure 4 there is an inscription in German.
In mathematics, the letter O with a '/' running through it means empty set: remove it.
Author Response
- We thank the reviewer for this note. Considering that this set of revisions has a time constraint (editorial office gives 10 days), the authors will do their best to improve the results analysis.
- We clarified the age inclusion: “Patients were all in good health, on average 42.55 years old (SD 7.02), with genetically caused hair loss (Androgenetica alopecia, AGA) in hair growth stage according to Nor-wood III-IV (male test persons) or Ludwig- Level 1 (female subjects) qualified for inclusion (Figure 1).
- We thank the reviewers for this remark. Analyses of significance have been conducted and reported.
- We thank the reviewer for this note. These data points were analyzed by means of descriptive statistics and no significance claim was made but rather the delta of averages was reported.
- We thank the reviewer for bringing this up. Figures have been edited and captions improved for better quality and clarity.
- We appreciate this remark. There was a mix up of panels because the figure panels were submitted individually to the journal. Figure 3 has now been composed by the authors to avoid this issue.
- We thank the reviewer for this opportunity to clarify. The dashed line represents the average hair loss which is physiological according to the literature. The average hair loss in the study cohort after 6 months corresponds well with the mean hair loss rate in healthy patients which was found to be approximately 40 according to the literature.
- Thank you for pointing this out. All figures have been checked and updated. Figures 3 and 4 have been recomposed by the authors.
- Thank you for highlighting this mistake. The Letter has been removed.
Reviewer 2 Report
This manuscript by Dominik Thor and coauthors, "A NOVEL hair restoration Technology Counteracts androgenic 2 hair loss and Promotes hair growth in A Blinded Clinical Trial ". This is a useful attempt and will be helpful reference for many researchers in the similar field. The manuscript details some interesting results. Manuscript is well explained results are consistent, and I may recommend the manuscript publishable in this journal with few minor comments
Figures captions are poor it needs to be more informative
Figures quality and resolution need to be improved to make it readable for the reader.

Author Response
- We thank the reviewer for this attention to detail, we have added statistical remarks to the caption to make them more valuable to the reader.
- We thank the reviewer for this well-reasoned suggestion. There was a mix up of panels because the figure panels were submitted individually to the journal. All figures have been checked and updated. Figures 3 and 4 have been recomposed by the authors.
Round 2
Reviewer 1 Report
I am truly disappointed that the authors evidently think that statistics is a minor part of their paper, as what is written in the paragraph on statistical analysis in part is completely meaningless:
"......... Two-tailed Student's t test was used to analyze for significance and results were considered significant at p ≤ 0.05"
The previous sentence does not make sense in English.
What is significance analysis?
"Additionally means of descriptive were used."
The previous sentence does not make sense. It is probably meant: The mean was used as a summary statistic.
Add a measure of variability because in statistics, measures of position without measures of variability are meaningless.
The authors probably mean: the Kolmogorov-Smirnov test was used to test normality distribution.
The two-tailed Student's t-test was used to test whether there was a difference between the mean of......[insert which variables].
Significance is set at p< 0.05.
However, the authors know that to compare a period of three times requires an ANOVA for repeated measures and then a post-hoc test.
Unfortunately, I do not see the modified figures in the new manuscript.
In English the decimal part is separated from the integer part by the dot so change all p=0..... with p=0....
Choose an appropriate number of decimal places for the p value. If p is less than 0.001 then write p<0.001.
I invite the authors to have a statistician re-read the manuscript and make the statistical part adequate and meaningful.
Moreover, it is still not clear in this article which variables are being studied, so I propose I suggest the authors remove the uninformative figures such as figure 2 and insert a table showing the pre- and post-values described as the mean and standard deviation and relative p-value.
Figure 2 in particular does not tell us what the initial and final value is, but only by how much it has increased.
at line 192 remove redundant concepts: Results were considered significant at p ≤ 0.05.
I think this paper is still a big mess, so I propose more revisions and the help of a professional statistician.
Author Response
We thank the reviewer for the insightful comments which helped to improve our manuscript.
We followed the reviewers suggestions and employed two independent statistical services in the US and the EU respectively to review our manuscript and the underlying dataset.
The reviewer indicated that the section labeled ‘Statistical analysis’ was confusing in several places. This section has been revised and expanded to clearly explain how the data were analyzed. Specifically, we note that five characteristics were compared before and after the use of Tomorrowlabs [HSF] hair restoration. For four of the characteristics – thickness, hair density, shine and elasticity, and hair growth – the follow-up period was nine months. For hair loss, the follow-up period was six months. We also note that the Shapiro-Wilk test was used to determine the normality of the distributions for the five characteristics. All distributions were normal with the exception of hair loss. Because only two points in time were being compared, paired t-tests were used for the data that were normally distributed – thickness, hair density, shine and elasticity, and hair growth. The Wilcoxon signed-rank test was used for the before/after comparison of hair loss which was non-normally distributed. We also indicate that we considered p-values less than 0.05 to be significant.
The reviewer suggested using an appropriate number of decimal places when reporting p-values (instead of using the exact values). We have made these changes.
The reviewer also suggested adding a table of variables that included before and after means, standard deviations, and p-values that correspond to tests of mean differences. We have included this (see Table 1).
We further rearranged figures 3 and 4 and included them directly into the revised manuscript.